# Probiotics in Citrus Fruits Products: Health Benefits and Future Trends for the Production of Functional Foods—A Bibliometric Review

**DOI:** 10.3390/foods11091299

**Published:** 2022-04-29

**Authors:** Shênia Santos Monteiro, Verônica Macário de Oliveira, Matheus Augusto de Bittencourt Pasquali

**Affiliations:** 1Programa de Pós-Graduação em Engenharia e Gestão de Recursos Naturais, Federal University of Campina Grande, Campina Grande 58429-900, PB, Brazil; shenia-monteiro@hotmail.com; 2Programa de Pós-Graduação em Administração, Federal University of Campina Grande, Campina Grande 58429-900, PB, Brazil; veronica.macario@ufcg.edu.br; 3Programa de Pós-Graduação em Bioquímica, Federal University of Rio Grande do Norte, Natal 59078-970, RN, Brazil

**Keywords:** citrus fruits, functional foods, health, probiotics

## Abstract

The relationship between food and human health drives the search for knowledge of food components that are related to these benefits. The scientific community shows a growing interest in the knowledge of the interactions between components of citrus fruits and probiotics to develop ways to improve the quality of the food produced. In this bibliometric review, a study of scientific publications is carried out on the potential of probiotics in citrus fermentation, addressing the importance and future trends of plant-based products in the functional food group as an alternative to the dairy market. The review process of the articles initially took place with a bibliometric analysis and was followed by a literature review. The Scopus database was used in the search for articles, carried out in May 2021. The use of foods as carriers of probiotics is an alternative that has been growing and the surveys evaluated show the desire to diversify the probiotics available on the market. In addition, it was observed that citrus fruits have great potential for the development of functional foods due to their high acceptability and possibilities of development and application in various products.

## 1. Introduction

Over the past two decades, numerous studies have redefined our understanding of microbes and changed our perception of the beneficial effects of microbes on human health [1]. Health-beneficial microorganisms are called probiotics, and to achieve health benefits they must be provided in adequate amounts. Probiotics are a kind of microecological regulator of humans or animals, which has the physiological function of improving intestinal health, promoting digestion and absorption, participating in immune regulation, alleviating allergic, antitumor, and lowering serum cholesterol [2]. They reach the consumer as fermentable [3,4] and non-fermentable [5,6,7] food products, in the form of powder [8,9,10], tablets [11,12,13], and capsules [14,15,16,17].

The enormous relevance currently dedicated to the importance of the intestinal microbiota in health and well-being has led to robust demand for probiotic foods, especially by younger generations who demand innovative products with better taste and flavor enrichments [18,19]. Human health and well-being also include adding more fruits and fruit derivatives to diets, driving the development of innovative fruit-based products [20].

Fruit juices, especially citrus fruits, are a promising carrier of probiotics thanks to the ability of lactic acid bacteria to also ferment slightly acidic plant and vegetable-derived substrates [6,21,22,23,24]. In addition, citrus fruits are popular around the world and used in a variety of preparations due to their unique sweet and moderately sour flavor profile, such as oranges [22]. However, some challenges are faced by the food industry when it comes to the development of probiotic foods and the addition of probiotics to citrus fruits because for these products it is very important to maintain the viability of the microorganism throughout the food storage time, which is a major challenge due to the sensitivity of probiotics to common processing conditions, such as heat treatment, low pH environments, high osmotic pressure, salt concentration, presence of dissolved oxygen, and high redox potential [25,26,27], which requires the design of effective barriers of physicochemical for the exploration of new foods.

Since information on the effects of adding probiotics to citrus fruit juices and the health benefits of these foods still requires extensive investigation, additional information should be gathered from existing research, especially on the influence of process conditions and presence of bioactive components of fruits and metabolites of probiotics in the fermentation of citrus fruit juices. Therefore, bibliometric analysis on accurate databases can show patterns and trends in scientific research around the world [28]. Considered an essential methodology for conducting research [29], this statistical tool has been increasingly used as a research performance evaluation tool, also being used to obtain information on citations and research results from countries, institutions, journals, and authors [20,30,31]. This methodology has achieved considerable interest as a quantitative analysis tool providing a comprehensive view of the growth and development of a specific research field such as food science [28,32,33], which leads to an understanding of future publishing trends based on available information on probiotics in citrus fruits. It is in this context that the purpose of this article was defined. Therefore, the objective was to carry out a mixed review, adopting a bibliometric analysis together with a literature review, to present information from the research and observe future trends for the production of probiotic foods based on citrus fruits.

## 2. Research Summary and Bibliometrics

To carry out this study, a mixed methodology was used, which consists of a bibliometric analysis followed by a literature review [34]. Bibliometric analysis refers to the combination of different structures, tools, and methods to study and analyze the literature and provide statistical analysis that summarizes research publications [35]. The documents used in the bibliometric analysis were obtained from a survey carried out in the Scopus database, conducted in May 2021. The Scopus database was chosen because it is the largest database of peer-reviewed abstracts and citations of literature, with bibliometric tools to monitor, analyze and visualize the research. The term ‘probiotics in citrus fruits’ was added to the item that includes title, abstract, and keywords in the Scopus database, obtaining 70 results. To refine the research, the types of documents were considered: articles and reviews; the following areas of study were also excluded: energy, environmental science, multidisciplinary economics, econometrics, and finance. All documents registered over the years in the Scopus database were considered for this research. After the restrictions were carried out, a result of 63 documents was obtained. Of the total documents considered, 56 are articles and 7 reviews, all in the final publication stage.

Bibliometric graphic mapping and data network visualization were analyzed using the VOSviewer software (Java version 1.8.0_291) (www.vosviewer.com accessed on 20 December 2021). The strategies adopted for scientometric mapping were carried out in two stages, which initially constitute a set of descriptive information on the topic, followed by the stage that portrays the collaborations of authors, keywords, co-occurrences or country links in the form of networks, the strength of the network link, density visualization, as well as heat maps, thus revealing information within a search field over time [36]. A performance analysis was carried out to discover a general pattern of research and estimate the global impact of a publication, including behavioral changes from scientific research on probiotics in citrus fruits [30]. The citation analysis and the analysis of the co-authorship of the obtained data were carried out considering the strength of the link by the program. In addition, based on keywords matches and citations, cluster analysis allowed the assessment of future trends for citrus-based probiotic foods.

To complement this information, the article presents a literature review on functional foods, the health benefits of citrus fruits and probiotics, and the challenges faced by the industry for the production of probiotic foods with citrus fruits.

## 3. Bibliometric Analysis

### 3.1. Evolution of Scientific Publication: Year, Countries, Organizations, and Publication Areas

A total of 63 documents were analyzed, of which 59 were published in English and the rest in Chinese, Korean, Russian, and Spanish. Figure 1 shows the numbers of publications registered on probiotics and citrus fruits in the Scopus database.

In Figure 1, when analyzing the data presented, it is observed that the first scientific publication on this subject, registered in the Scopus database, was in 2006, as the registration of 1 article. In the following year, no publications were registered. In the years after 2007, the number of published works ranged from 1 to 7 publications per year, observing the peak of publications in the year 2020, with 12 publications, of which 5 of the 12 publications are scientific articles that investigated the relationship between probiotics and orange juice and its health benefits.

The growth in research on probiotics in citrus fruits reflects the demand for functional foods. The world population has been interested in diets that promote a significant increase in well-being [37] and fruits are a perfect combination of flavor and nutritional compounds, fundamental sources of vitamins (vitamin C and group B), provitamin A, dietary fiber, minerals, and phytochemicals of importance in the human diet [7,24,38]. Specifically, within the functional foods group, probiotics grow given the potential health benefits attributed to viable cells and also to fermentation products. Such benefits of probiotics added to the natural source of bioactive compounds in citrus fruits demand studies to understand the interactions between these components for the development of functional food products [39,40,41,42,43].

Brazil stands out with the largest number of publications related to probiotics and citrus fruits (Table 1), from the year 2016 to May 2021. Spain and China occupy second and third place, with 11 and 8 publications, respectively, highlighting as a source, the titles of periodicals: LWT—Food Science and Technology, with 11 publications; Food Research International, with 5 publications and Food Chemistry, with 3 publications.

Our results show that the nations with the highest number of publications also have their authors among the most cited articles, as do the United Kingdom, China, and Spain. The predominance of Brazil in the number of publications related to probiotics in citrus fruits may be related to the abundance of citrus fruit varieties and the importance of the Brazilian citrus industry, world leader, responsible for more than 80% of world exports of orange juice and more than 30% of all world production of the fruit, according to the CNA (Confederation of Agriculture and Livestock of Brazil).

Among the affiliations of authors and co-authors, it is noted when analyzing Table 1 that the University of Reading registers the highest number, with 4 publications. Of the 10 institutional organizations mentioned in Table 1, 4 are Brazilian institutions that together add up to 8 publications.

The scientific papers that addressed the subject ‘probiotics in citrus fruits’ were published in 9 thematic areas (Table 1). Of the scientific works analyzed, around 73.02% were published in the area of Agriculture and Biological Sciences, followed by the area of Medicine.

The area of agriculture and biological sciences involves the research of food matrices as a vehicle for delivering probiotics. The numbers presented here highlight the importance of studying probiotic foods, especially the use of citrus fruits for the development of new functional foods with quality and food safety.

Allied to the commitment to the development of products that facilitate the delivery of probiotics to the consumer, such as citrus fruit-based foods, the concern with health drives scientific research on probiotics, and their potential health benefits in the medical field.

### 3.2. An Analysis of the Reference Data

In an analysis of citations and future trends based on the documents evaluated, the participation of 38 countries in the construction of scientific knowledge on issues related to probiotics in citrus fruits was observed. In Figure 2, we observe the collaboration network of the most cited countries until May 2021, highlighting the United Kingdom and Spain as the most cited countries in this theme; this may be related to publications registered the longest. Brazil has emerged in this scenario since 2016 and only 2 publications were registered in the last months. The collaboration of these countries tends to grow, especially in the field related to the production of foods with added probiotics.

A quotation of the most cited journals and authors was carried out, and the results showed that the most cited journals were: LWT—Food Science and Technology, International Journal of Food Microbiology and Food Chemistry, recording 168, 125, and 89 citations in scientific publications, respectively. This result shows a list of the best journals to publish articles in the area of probiotic foods based on citrus fruits, which reflects the impact of citation and the growth of journals since it is noted that the most cited articles were published in journals of high impact, considered popular in the food science field. In this analysis, journals with a minimum registration of 2 documents were considered.

Of the authors with more mentions, the following stand out Charalampopoulos, D., Nualkaekul, S., Barão, C.E., and Magnani, M., respectively. Nualkael. S. in partnership with Charalamapopoulos, D. in their contributions to scientific research investigated, in these analyzed publications, the survival of probiotic bacteria in acidic solutions such as citrus fruit juices [44,45]. Barão, C.E. contributed to the study of the impacts of the addition of probiotics on the quality of juices produced with orange [6,46]. Magnani, M., in his contributions to the works analyzed in this study, carried out a study of the impacts of unit operations on the survival of probiotic strains [47] and the survival of probiotic strains during frozen storage and when incorporated into the apple, orange, and grape juice [48]. Orange stands out among citrus fruits as a potential vehicle for probiotics, as can be seen in the aforementioned studies.

The analysis of the occurrence of keywords in the selected studies was used to identify current trends in scientific research on the topic of probiotics in citrus fruits. In this study, all keywords (own and indexed words) were analyzed. In Figure 3, it is possible to observe a network of keywords where the terms located nearby are the ones with the highest registration in the evaluated data. Still, in Figure 3, it can be observed over the years that the most recently observed terms are related to the production of food for the benefit of human health. In addition to knowledge about probiotic strains, ability to survive the passage through the gastrointestinal tract, and the beneficial functions of the host, over time, the search for knowledge about the products of fermentation of plant bases and techniques to guarantee the delivery of products is observed with high quality and food safety.

The study of keywords identified 4 linked groups, 3 of which were more evident. The first group collects the greatest number of keywords. In this group, the relationship between probiotic bacteria in fruit juice and citrus fruits is observed from the perspective of conservation, with the words microencapsulation and food storage highlighted. This group of keywords shows that investigations into the survival of probiotic bacteria in acidic solutions such as fruit juices have been accompanied by the search for conservation and storage methods that ensure that the product’s viability reaches the consumer. The second group is strongly linked to the first group of keywords. In this group, strong links were observed between the word’s fermentation, probiotics, citrus fruits, and chemistry. In this group, the importance of knowing the interference of the chemical composition of fruits and vegetables and the changes in chemical components during fermentation to obtain new products is highlighted. While the third group of keywords makes a connection between probiotic bacteria, citrus fruits and their relationship with the benefits to humans, highlighting the words metabolism and supplemental diets in publications carried out in the last five years (2017–May 2021).

In Figure 3, the most cited keywords in the evaluated scientific publications are highlighted. By analyzing Figure 3, the evolution of keywords over the years is noted, revealing a trend towards researching probiotic products in citrus fruits linked to supplemental diets, the study of metabolisms and delivery methods of bioactive compounds, and their beneficial potential for the health of man.

## 4. Functional Foods

Functional foods emerge among food products in the scenario where the growth of diseases caused by poor diet is observed. Functional foods, in addition to nourishing and supplying the body’s needs, contain components that are related to various health benefits; these benefits are related to well-being and/or the ability to reduce or prevent diseases such as obesity, high cholesterol, diabetes, gastrointestinal and other diseases.

### 4.1. Beneficial Health Potential of Citrus Fruits

Epidemiological studies suggest that diets rich in fruits and vegetables are related to a lower incidence of various chronic diseases, as well as to the prevention and delay of age-related diseases [49,50]. The importance of citrus fruits for human nutrition is due to the presence of components such as vitamins and biologically active phytochemicals that provide antioxidant benefits, in addition to nutritional components such as carbohydrates, dietary fibers, and minerals [51].

Citrus fruits belonging to the genus Citrus (*F. Rutaceae*) are of various shapes and sizes, commonly known as oranges, tangerines, limes, lemons, grapefruit, and ciders [52]. In addition to the edible fraction of the fruit, waste parts such as the skin have a wide variety of secondary components, especially flavonoids with substantial biological activities compared to other parts of the fruit [53].

Citrus flavonoids represent the main class of secondary metabolites in citrus fruits and are considered the main category of dietary polyphenols. Bioflavonoids are gaining more attention in cancer treatment for their strong antioxidant and bioactive potential [54]. Widely found in several citrus fruits such as oranges, lemons, and other vegetables and fruits, flavonoids, such as hesperidin, are recognized as a potential phytocompound to which numerous health benefits are attributed, including antimicrobial, anticancer, antioxidant, anti-inflammatory, antidiabetic, and cardiovascular protective activities, especially chemopreventive and chemotherapy potential [55,56,57,58].

Promising candidates for prebiotics, non-digestible oligosaccharides derived from citrus fruit by-products, such as lemon peel, are included in the concept of functional foods and through fermentation by beneficial bacteria such as *Lactobacillus* and *Bifidobacterium* genus, and thus confer physiological effects. positive for the organism and the health of the host [59,60,61,62]. An example of the prebiotic potential of pectic oligosaccharides (POS) derived from lemon peel by-products was discussed by Gómez et al. [59] where it was observed that the POS complex derived from by-products such as lemon peel can be used as a carbohydrate source by *Lactobacillus* and *Bifidobacterium*, showing that the low molecular weight POS obtained from lemon peel has a specific bifidogenic effect for *B. lactis*.

Citrus fruits and their by-products have a wide technological potential for the use of their components in the cosmetic, pharmaceutical, and food industries. In agribusinesses, fruit by-products are still often considered as waste; however, these materials such as peel and seeds have a wide variety of metabolites of interest, as mentioned above, due to the relationship of these metabolites with several benefits to consumer health, which has driven the scientific community to describe these beneficial relationships and the technological potential of citrus fruits and their by-products.

### 4.2. Challenges of Probiotics in Citrus Fruits

As probiotics must remain viable during processing, storage, and passage through the gastrointestinal tract, the development of probiotic food products faces several challenges. These challenges are related to the stress triggered by unit operations in processing; deficiency in substrate composition, limiting the growth of more demanding probiotic species; and sensory changes.

The way of preparing probiotic foods is also a challenge for the food industry. In fruit juices, probiotic cultures can be added as biomass or by direct addition of freeze-dried culture [63]. However, new methods of adding probiotics to foods have been explored to overcome disadvantages, such as the need for specific equipment and specialized operators, in addition, all these operations can compromise the viability of the culture and the sensory properties of the food. An example is the addition of probiotics through application in a coating [64,65,66].

Microencapsulation has shown to be a promising strategy to protect probiotic cultures from adverse conditions found in citrus fruit derivatives because, despite the high nutritional value of citrus fruits, their low pH and insufficient amount of some peptides and amino acids are limiting factors in the preparation of products citrus fruit probiotics [67].

Microencapsulation is a process that has been widely investigated for the production of probiotic microcapsules. The microcapsules consist of a core containing active ingredients and a polymeric shell, acting as a protective membrane capable of regulating the exchange of substances in and out [68]. Microencapsulation techniques are grounds for extensive research aimed at optimizing the process and preserving the viability of the probiotic culture. Choosing the proper method of preparation of microcapsules depends on several factors, such as physicochemical properties of the selected materials, desired particle size, and release pattern [63].

The production of microcapsules containing *Lactobacillus acidophilus* LA-02 using the complex coacervation technique followed by crosslinking with transglutaminase for addition to different fruit juices showed that the microcapsules had a protective effect on *L. acidophilus* ensuring the viability of the probiotic [67]. This study shows that the complex coacervation technique is a viable alternative to overcome the challenges of probiotic production in citrus fruits.

The extrusion method is a simple and low-cost encapsulation technique compared to other encapsulation methods. The materials used are essential to guarantee the viability of the probiotics in the food matrix and during the digestion process in the gastrointestinal tract. In the encapsulation of probiotics, alginate is a material commonly used, but it has some disadvantages that, when used in mixtures, can present greater efficiency in the protection of probiotics. The mixture of alginate with Persian gum with fructooligosaccharides and inulin in the microencapsulation of *L. lactis* ABRIINW-N1 by the extrusion method showed surprising results, with high encapsulation efficiency keeping the cells viable in the orange juice stored at 4 °C [69].

Another method of encapsulating probiotics with potential is the layer-by-layer strategy. Another method of encapsulating potential probiotics is the layer-by-layer strategy. The encapsulation of the probiotic *Bacillus coagulans* using the strategy layer by layer using chitosan-alginate through electrostatic interactions showed advantageous results about the survival of the probiotic against damage resulting from adverse conditions of the human digestive system. This method is an alternative with promising growth for application in the development of probiotic products based on citrus fruits since positive results have been observed, guaranteeing the viability of probiotic cells in the digestive system [19,70,71].

In addition to feasibility, studies show a concern with the sensory and nutritional properties of fruit juices with added probiotics. In this scenario, microencapsulation has a special function. Bonaccorso et al. [63] in a study of the encapsulation of the *Lacticaseibacillus rhamnosus* GG strain in an alginate system, carried out through the ionotropic gelling technology and application of microcapsules in orange juice, it was found that the encapsulated bacteria did not affect the macroscopic properties or the microbiological characteristics of orange juice.

Another way to add probiotics and their metabolites to citrus-based foods is through fermentation. In the same way that bacteria have beneficial effects on consumer health, some foods fermented by lactic acid bacteria can bring health benefits due to the permanence of viable cells and also the components produced during fermentation. These beneficial effects of fermented foods include improved gut microflora [72] and perennial allergic rhinitis [73], as well as lower abdominal adiposity [74] in humans [22].

On the other hand, fermentation can bring changes in flavor to the product. Such changes in sensory properties, for example, in citrus fruit juices may be desirable when well accepted by consumers or undesirable [75]. Some studies sought to investigate changes in chemical composition and sensory properties in juices fermented by lactic acid bacteria. Yuasa et al. [22] investigated the viable cell count, chemical composition, and sensory properties of three citrus fruit juices fermented by *L. plantarum* SI-1 and *L. pentosus* UM-1 compared to their unfermented parts and observed that citrus juices fermented had enough viable cells to bring health benefits and no difference was observed in the sensory properties of the juices by fermentation. In passion fruit fermented with *Lactiplantibacillus plantarum* CCMA 0743 as single culture and co-culture, the sensory profile of the juice was modified by single and co-culture fermentation [4].

## 5. Health Benefits of Probiotic Foods

New beneficial functions attributed to probiotics and citrus fruit components are being discovered all the time. The consumption of probiotics with natural bioactive compounds is a “green” alternative to medicines to maintain blood pressure control, good mental health, boost immunity, eliminate sleep disturbances and other health problems, especially in the current situation of the COVID outbreak-19 [14].

When it comes to the health benefits of probiotics, the beneficial role of this microorganism has already been reported in helping to treat and regulate various diseases. The action of probiotics involves strengthening the barrier function of the intestinal mucosa, improving the balance of intestinal bacteria, promoting the growth of beneficial bacteria, inhibiting pathogen growth, increasing immunity, and so on [76].

Another aspect of the health benefits of probiotics is the biological detoxification of chemical food contamination. Environmental and chemical pollutants produced by industry and agriculture, known as food contaminants and called xenobiotics, have long-term negative effects on human health [77]. In this context, probiotics appear as an effective tool to prevent dysbiosis induced by foreign contaminants and alleviate toxicity [77].

In addition, the concepts of paraprobiotics and postbiotics are still a little-explored universe and of great relevance for the production of probiotic foods in citrus fruits. The paraprobiotics are inactivated (non-viable) microorganisms that benefit consumers’ health. Otherwise, postbiotics are defined as soluble factors (metabolic by-products or by-products) secreted by live bacteria or released after bacterial lysis, which can benefit the host [78,79]. Therefore, applications of cellular components or metabolites derived from probiotic strains are gaining more interest [79].

The concept of paraprobiotics was proposed to indicate the use of inactive (non-viable) microbial cells or cell fractions that confer benefits to the health of consumers [79,80,81,82,83,84,85,86,87,88,89,90]. Postbiotics, on the other hand, is defined as soluble factors (metabolic products or by-products) secreted by live bacteria or released after bacterial lysis, which can benefit the host [78,79].

The mechanism of action of paraprobiotics is not fully understood, but it is known that they are capable of acting in immunomodulation [91]. Several studies suggest that these immunomodulatory effects are produced by different components of the microbial cell [92]. These components are, for example, β-glycans, chitin, mannoproteins, teichoic and lipoteichoic acids [93], cell homogenates [94], peptidoglycans (PGN), lipopolysaccharides (LPS), and DNA [95]. Some beneficial effects of postbiotic paraprobiotics produced by probiotic microorganisms are summarized in Table 2.

## 6. Future Trends of Probiotic Functional Foods

The research carried out so far on the beneficial potential of the use of probiotics in human food has contributed a lot to the health sector and the food industry. However, many challenges still exist around the production of safe probiotic foods. First, knowing the relationships of probiotic microorganisms with human health contributed to leveraging studies in this area and creating processing technologies that ensure the safe and viable delivery of probiotics so that health benefits are achieved. Although food is a promising vehicle for the supply of probiotic bacteria, much remains to be known about the interactions between these microorganisms and food and so on in human health. The survival of probiotics in the gastrointestinal tract is essential for modulating the intestinal microbiota and the immune system, helping to prevent diseases such as the respiratory system [103].

Encapsulation and microencapsulation techniques well-researched techniques to ensure the delivery of the probiotic to the place of interest. However, the food matrix and the process conditions are of growing interest by the scientific community, since the properties of each product are specific, requiring continuous research to optimize the process and knowledge of the optimal conditions.

On the other hand, the interest in knowing the relationships between paraprobiotics and postbiotics in food and health emerges in the production of functional probiotic foods. The metabolites or products of the action of these microorganisms are of interest to the food and pharmaceutical industry since it has a great potential for the development of innovative products that meet the consumer’s needs for quality food and bring benefits to the health.

## 7. Study Limitations

Some limitations of this study should be considered when evaluating the results. First, we only used the Scopus database to retrieve the relevant literature; thus, publications in journals not indexed in Scopus were not considered. The Scopus database was used in this study because it has a relevant scope on the subject and also because it is considered an accurate, reliable, and frequently used database for bibliometric analyses [28]. Another important point is to select only documents of the research and review article type, not being considered conference documents, book chapters, and others. Despite the limitations presented, the results presented here are considered relevant to promote discussions and direct the development of scientific knowledge about probiotics in citrus fruits.

## 8. Conclusions

We apply a mixed methodology that includes a scientometric analysis and literature review to provide future perspectives for the development of probiotic products using citrus fruits.

We observed that the results indicate the growth of scientific publications related to the topic, but we identified that most publications are about investigations of the health benefits attributed to probiotics. At the same time, citrus fruits appear to be a promising plant base for the development of probiotic foods. This has been especially attributed to the high acceptance of citrus fruit derivatives by the world population, and also the source composition of essential nutrients for the human diet. Another relevant aspect observed in the results of this study is the scarcity of studies directed to the sensory characteristics of citrus fruit products added with probiotics. The bibliometric analysis showed that little is known about the interactions in the fermentation of citrus juices with probiotics and sensory responses, as well as the effect of including these products in the human diet. Thus, the citrus fruit juice fermentation process represents an opportunity to identify new products composed of fermentation by probiotic microorganisms and their health benefits and potential use in the food industry.

In this way, we identified directions for future research for the development of probiotic foods in citrus fruits, as well as the main collaborating countries and leading journals in the publication of scientific articles on the subject. These journals have a relevant academic impact and thus constitute an important vehicle to promote studies for the development of functional foods and the use of emerging technologies.

## Figures and Tables

**Figure 1 foods-11-01299-f001:**
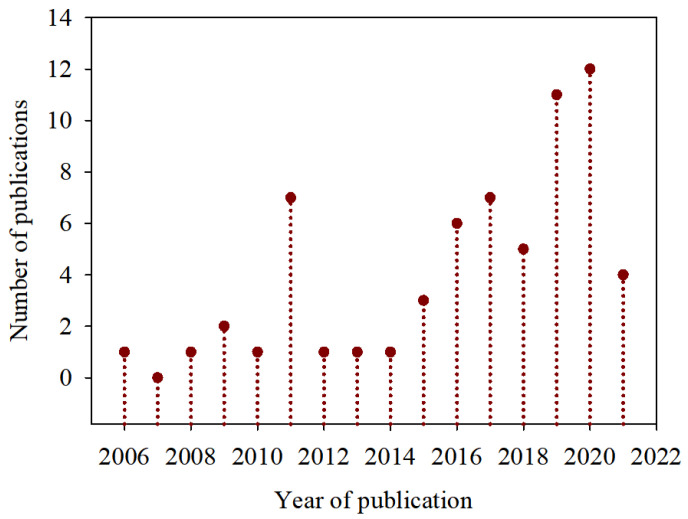
A number of scientific publications on probiotics in citrus fruits over the years (research carried out in the Scopus database in May 2021).

**Figure 2 foods-11-01299-f002:**
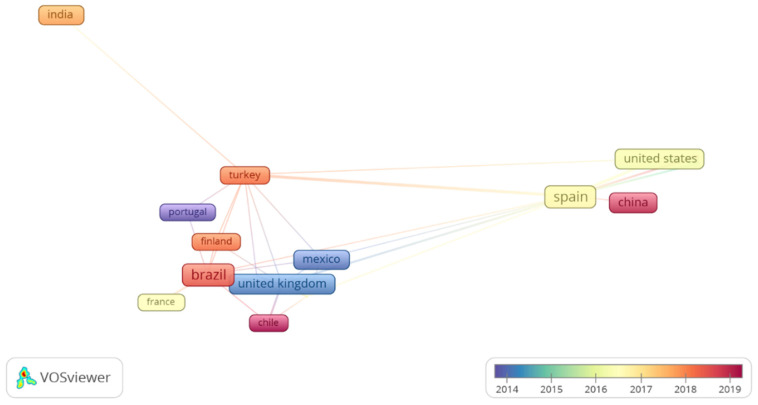
Visualization map of the collaboration network of countries according to co-occurrence over the years. Each frame represents a country. The frame size indicates its frequency of occurrence. The count of citations received by articles that contain the term or phrase by year is represented according to a color scale. The closest frames indicate the terms that co-occurred most frequently over the years.

**Figure 3 foods-11-01299-f003:**
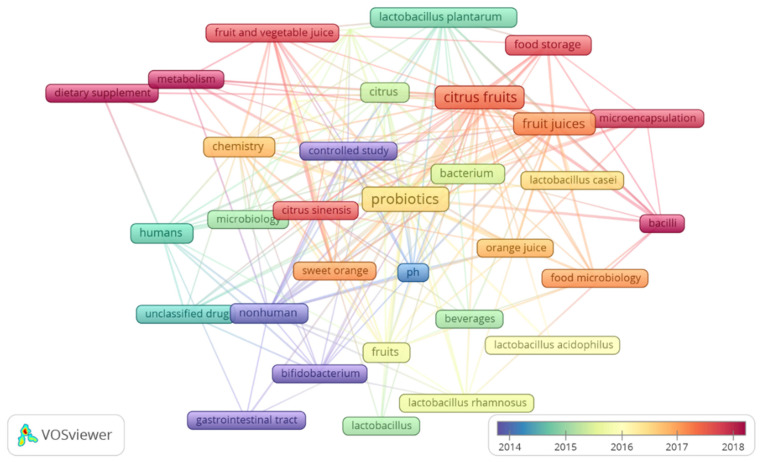
Keyword network visualization map according to co-occurrence. Each frame represents the keywords. The frame size indicates its frequency of occurrence. The count of citations received by articles that contain the term or phrase by year is represented according to a color scale. The closest frames indicate the terms that co-occurred most frequently over the years.

**Table 1 foods-11-01299-t001:** Rankling of the main contributors to the scientific production of probiotics in citrus fruits about the number of publications by country, institutional affiliation, and publication area.

Ranking	Name	Number of Publications	Percentage (%) *
	**Country**		
1	Brazil	12	19.05
2	Spain	11	17.46
3	China	8	12.70
4	United States	6	9.52
5	United Kingdom	6	9.52
6	India	5	7.94
7	Iran	4	6.35
8	Italy	4	6.35
9	Mexico	4	6.35
10	Poland	3	4.76
	**Organizations**		
1	University of Reading	4	6.35
2	Alma Mater Studiorum Università di Bologna	3	4.76
3	CSIC-Instituto de Agroquimica y Tecnologia de los Alimentos IATA	2	3.17
4	Jiangnan University	2	3.17
5	Universidade de São Paulo—USP	2	3.17
6	Universidade Federal da Paraíba—UFPB	2	3.17
7	Universitat Politècnica de València	2	3.17
8	University of California, Davis	2	3.17
9	Universidade Federal do Ceará	2	3.17
10	Universidade Estadual de Maringa	2	3.17
	**Publication areas**		
1	Agricultural and Biological Sciences	46	73.02
2	Medicine	11	17.46
3	Immunology and Microbiology	8	12.70
4	Nursing	8	12.70
5	Biochemistry, Genetics and Molecular Biology	7	11.11
6	Chemistry	7	11.11
7	Chemical Engineering	4	6.35
8	Pharmacology, Toxicology and Pharmaceutics	3	4.76
9	Engineering	2	3.17

* (%): Percentage of 63 publications.

**Table 2 foods-11-01299-t002:** Health benefits of paraprobiotics and postbiotics.

Probiotic	Heath Benefits	References
*L. rhamnosus* GG	Anti-inflammatory	[78,96]
*L. brevis* SBC8803	Anti-inflammatory and improved permeability of the epithelial barrier, recovery from intestinal injuries	[97]
Fermented milk containing *L. gasseri* CP2305	Regulation of bowel function in patients with tendencies to constipation	[98]
*L. casei* B1	Anti-oxidative, anti-proliferative, and anti-adhesion activity against *S. aureus*	[99]
*L. acidophilus* DDS-1	Increased levels of short-chain fatty acids (butyrate, propionate, and acetate)	[100]
*L. plantarum*	Antibiofilm activity against *S. mutans*	[101]
*L. fermentum* S1	Anti-oxidative and anti-biofilm effect against *E. coli* and *S. aureus*	[102]

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
