# Peer review of "Probiotics in Citrus Fruits Products: Health Benefits and Future Trends for the Production of Functional Foods—A Bibliometric Review"

_foods, 2022, doi:10.3390/foods11091299_

Round 1

Reviewer 1 Report

1-Similarity 41%  is not acceptable. Regarding the attached file, many sentences have been copied and paste from the the following two references:

Anely Maciel de Melo, Francisco Lucas Chaves Almeida, Atacy Maciel de Melo Cavalcante, Mônica Ikeda et al. "Garcinia brasiliensis fruits and its by-products: Antioxidant activity, health effects and future food industry trends – A bibliometric review", Trends in Food Science & Technology, 2021

Crossref Siegried Lillo-Pérez, María Guerra-Valle, Patricio Orellana-Palma, Guillermo Petzold. "Probiotics in fruit and vegetable matrices: Opportunities for nondairy consumers", LWT, 2021

2- Some statements need references: e.g. page 2 First sentence:

Fruit juices, especially citrus fruits, are a promising carrier of probiotics.

3- Cross references is a big mistake. Please avide using a second reference instead of citation to main one.

Just for instant: The reference number 6 is:

Soukoulis, C.; Behboudi-Jobbehdar, S.; Macnaughtan, W.; Parmenter, C.; Fisk, I.D. Stability of Lactobacillus Rhamnosus GG Incorporated in Edible Films: Impact of Anionic Biopolymers and Whey Protein Concentrate. Food Hydrocolloids 2017, 70, 345–355, doi:10.1016/j.foodhyd.2017.04.014.

Is not a good reference for following paragraph:

Fruit juices, especially citrus fruits, are a promising carrier of probiotics. However, some challenges are faced by the food industry when it comes to the development of probiotic foods and the addition of probiotics to citrus fruits, due to the sensitivity of probiotics to common processing conditions, such as heat treatment, low pH environments, high osmotic pressure, and high redox potential. Thus, the study and design of effective physicochemical barriers to stabilize organisms are essential for their full commercial exploitation in a wide variety of foods [6].

5- Table 1 is not enough informative. The tables and figures should sum,arize lots of information independent from the text. Legend of first Table is not clear and informative:

Table 1. Scientific publications by country, institutional affiliation, and publication area.

6- Chanelneges of using citrus fruit as carrier of probiotics has not been discussed.

7- Conclusion is not informative and deep enough. Many of sentences of this part was clear before this study:

The rise of functional probiotic foods provoked the scientific community to research the relationship between the use of these micro-organisms and their relationship to human health. Food has shown great potential for the delivery of these components to consumers, however, the demand for healthier foods together with the search for foods with low cholesterol levels or that meet the public with food restrictions such as lactose intolerant or culture adept vegan provoked the food industry to seek alternatives to meet consumer needs. The use of fruits and vegetables is a promising alternative for the development of innovative probiotic functional foods. Bibliometric analysis showed that little is known about the interactions in citrus juice fermentation with probiotics and sensory responses, as well as the effect of including these products as part of the human diet. Thus, the citrus fruit juice fermentation process represents an opportunity to identify new compounds products of fermentation by probiotic micro-organisms and their health benefits and potential use in the food industry

8- Gap of research has not been discussed.

9- there are many similar paper not cited in the text.

attached please find the comments and also plagiarism detection.

Reviewer 2 Report

Manuscript ID: foods-1540878

Title: Probiotics in citrus fruits: health benefits and future trends for the production of functional foods - A bibliometric review

Authors: Shênia Santos Monteiro , Verônica Macario de Oliveira , Matheus Augusto de Bittencourt Pasquali

Overview and general recommendation:

The aim of the manuscript was a mixed review, adopting a bibliometric analysis together with a literature review on probiotic foods based on citrus fruits.

Despite the fact that the topic of the manuscript is interesting, certainly current and fits into one of the leading trends in food, i.e. pro-health food, including probiotics, in my opinion the manuscript needs improvement.

Below I give some concerns that require review:

Major comments

  1. According to the title, the expected subject of the manuscript should be related to probiotics used in products obtained from citrus fruits (such as juices or by-products) or in the fruits themselves. However, some of the manuscript (chapters "Health benefits of probiotic foods" or "Future trends of probiotic functional foods") concerns very general, commonly known information about probiotics.
  2. When reading the manuscript, one can get the impression that the authors in the first part - chapter Bibliometric Analysis, used different references than to describe chapter 4. Functional Foods, and therefore this second part deals with general topics, less related to the combination of both probiotics and citrus fruits.

Mainor comments

Title – “Probiotics in citrus fruits” – title should be more precise, first impression might be, that some native microflora of citrus fruits has a probiotics properties, and according to the text it is not just fruits, but products like juices or by-products.

There is no line numerations – which makes it difficult to refer to specific information in the text

Introduction – please give more information for justifying the choice of the aim of the manuscript, why it is important to write whole article about probiotics in citrus fruits. There is a sentence: “Fruit juices, especially citrus fruits, are a promising carrier of probiotics.” – but it is more an information, than a hypothesis, and if so it needs a support in the references.

Figure 1 – “Evolution ….” - Is it an evolution? Evolution is a process of gradual changes, it is rather a list of publications by year of publication?

Bibliometric Analysis: “The growth in research on probiotics in citrus fruits reflects the demand for functional foods.” - Unfortunately, this information is somewhat overestimated. Are there any guidelines that indicate that the number of publications translates into consumer interest?

Bibliometric Analysis: “Spain and China occupy the second and third place, with 11 and 8 publications, respectively, highlighting as a source, the titles of periodicals: LWT - Food Science and Technology, with 11 publications; Food Research International, with 5 publications and Food Chemistry, with 3 publications.” – please conclude, what does this data show?

Bibliometric Analysis– “The highlight of Brazilian institutions in the subject of probiotics and citrus fruits is due to the importance of the Brazilian citrus industry…” - This explanation would better fit in the part of the text where Brazil's position is described as a leader in the number of publications.

Figure 2 – Title please use the capital letter at the beginning. Besides in text there is an information that it is a collaboration not just a list. How to interpret the colours, is the newest citation or collaboration in red colour??

Bibliometric Analysis – “A quotation of the most cited journals and authors was carried out, and the results showed that the most cited journals were: LWT - Food Science and Technology, International Journal of Food Microbiology and Food Chemistry, recording 168, 125, and 89 citations in scientific publications, respectively. In this analysis, journals with a minimum registration of 2 documents were considered”/- please add the conclusion, is this positive or negative?

Bibliometric Analysis – “Barão C. E. contributed to the study of the impacts of the addition…” - impacts on what? quality of juice or survival of the probiotics?

Figure 3 – please add the conclusion or comment on how the date change the fields of interests?

Latin names – pleas change all Latin names on italic

4.1. Beneficial health potential of citrus fruits - I do not fully agree with the very concept of chapter because it treats the subject fruit separately form probiotics, even if they do have prebiotic ingredients, but my concern is , if this information comes from articles selected for bibliometric testing, i.e. those containing and required search words (citrus fruits and probiotics)? In my opinion this and further part all should based on those publications that were previously described as selected?

4.2. Challenges of probiotics in citrus fruits “However, new methods of adding probiotics to foods have been explored to overcome disadvantages…” – what are the new methods?

Sentences based on references 44 and 46 - Are these statements based on the general literature or on citrus and probiotics?

4.2. Challenges of probiotics in citrus fruits -“…kiwi juice fermentation…” – is kiwi a citrus fruit?

  1. Health benefits of probiotic foods – “…has already been reported…” – twice used, according to the authors themselves, this subject has already been described upon in the literature, so it may not make sense to described it again?
